# Application of *Escherichia coli*-Derived Recombinant Human Bone Morphogenic Protein-2 to Unstable Spinal Fractures

**DOI:** 10.3390/bioengineering10101114

**Published:** 2023-09-22

**Authors:** Young-Hoon Kim, Jun-Seok Lee, Kee-Yong Ha, Sang-Il Kim, Ho-Young Jung, Geon-U Kim, Yongwon Joh, Hyung-Youl Park

**Affiliations:** 1Department of Orthopedic Surgery, Seoul St. Mary’s Hospital, College of Medicine, The Catholic University of Korea, Seoul 06591, Republic of Korea; boscoa@catholic.ac.kr (Y.-H.K.);; 2Department of Orthopedic Surgery, Eunpyeong St. Mary’s Hospital, College of Medicine, The Catholic University of Korea, Seoul 03312, Republic of Korea; 3Department of Orthopedic Surgery, Kyung Hee University Hospital at Gangdong, Seoul 05278, Republic of Korea

**Keywords:** unstable spinal fractures, recombinant human bone morphogenic protein-2, anabolic agents

## Abstract

(1) Background: Recently, *Escherichia coli*-derived recombinant human bone morphogenetic protein-2 (*E. coli*-derived rhBMP-2) has been increasingly applied to different types of spinal surgeries and reported to achieve successful fusion. This pilot study aimed to evaluate the clinical efficacy and safety of rhBMP-2 in patients undergoing posterior instrumented fusions for unstable spinal fractures. (2) Methods: This study included ten consecutive patients undergoing spinal surgery using *E. coli*-derived rhBMP-2 with more than one year of follow-up. Radiologic outcomes were compared, including the average fracture healing period, local kyphosis correction, and clinical outcomes between preoperative and the last follow-up. (3) Results: The average time of radiographic union was 99.9 ± 45.4 (62–192) days, with an average use of 5.2 ± 3.9 months of anabolic agents. Radiologic parameters such as anterior vertebral height and vertebral wedge angle were significantly corrected postoperatively and at the last follow-up. Clinical outcomes other than leg pain were significantly improved after the surgery. In addition, four patients with preoperative neurologic deficits showed improved neurologic status. (4) Conclusions: Combined with the anabolic agents, applying *E. coli*-derived rhBMP-2 to the fractured vertebral body could be an effective surgical treatment for unstable spinal fractures. Further trials are needed to validate this result.

## 1. Introduction

Spinal fractures cause substantial disability and decrease quality of life due to pain, deformity, and neurologic deficits. Most spinal fractures could be managed with conservative treatment such as orthosis and medications [1]. However, in some cases, surgical interventions are needed for unstable spinal fractures or neurologic deficits. Studies for surgical indications and techniques have been reported to achieve faster solid fusions for spinal fractures and prevent complications resulting from delayed union and pseudarthrosis. In addition, biomaterials and supportive osteoporosis medications have been developed and applied to spinal surgeries [2].

Bone morphogenetic proteins (BMPs) are multi-functional growth indispensable to various developmental processes, including cardiogenesis, neurogenesis, and osteogenesis. Among the BMPs, BMP-2 was the first characterized BMP and has been reported to play essential roles during embryonic development, as well as bone remodeling and homeostasis in adulthood [3]. Moreover, the osteoinductivity of BMP-2 has been evaluated extensively and reported to be a successful bone graft substitute [4,5]. Since the Food and Drug Administration approved the usage of recombinant human BMP-2 (rhBMP-2) for spinal fusion surgery, tibial shaft repair, and maxillary sinus reconstructive surgery, rhBMP-2 has been applied to many orthopedic surgeries as well as dental/maxillofacial field [3,5].

The human genes encoding BMP-2 can be transfected into two cell lines: Chinese hamster ovary (CHO) cells, and Escherichia coli cells [6]. Purifying rhBMP-2 through CHO cells requires a high cost in order to obtain a sufficient amount due to incomplete monomer processing and low yields [7,8]. On the other hand, the inferior effect of osteoblastic differentiation in mesenchymal stem cells and production in non-active aggregated form have been reported to be disadvantages of *Escherichia coli*-derived rhBMP-2 (*E. coli*-derived rhBMP-2) compared to CHO-derived rhBMP-2 [9]. However, recent studies have reported that *E. coli*-derived rhBMP-2 had biological activity equivalent to CHO-derived rhBMP-2 and the economic advantage with the production of large amounts and high purity by dimerization through biochemical processing [8,10].

Recently, *E. coli*-derived rhBMP-2 has been increasingly applied to different types of spinal surgeries and reported to achieve successful fusion [11,12,13]. However, to the best of our knowledge, the clinical efficacy of rhBMP-2 for vertebral fractures has yet to be reported. Therefore, this pilot study of *E. coli*-derived rhBMP-2 aimed to evaluate the clinical efficacy to enhance solid bony union and safety regarding rhBMP-2-related complications in patients undergoing posterior spinal fusions for unstable vertebral fractures for the first time.

## 2. Materials and Methods

### 2.1. Patients Undergoing Spinal Surgery

Patients undergoing posterior spinal fusion using *E. coli*-derived rhBMP-2 for unstable spinal fractures were included in this study. In this study, inclusion criteria were as follows. (1) patients undergoing spinal surgery for unstable spinal fractures or neurologic deficits regardless of the cause; (2) patients who agree with the use of rhBMP-2. Unstable spine fractures included three-column injuries such as a Chance fracture or flexion-distraction injury with posterior ligamentous complex injury. Burst fractures with neurologic deficits due to canal encroachment were also surgical indications [14].

The exclusion criteria were as follows. (1) Compression and burst fractures without neurologic deficits, which were treated conservatively; (2) patients with less than 1-year follow-up for evaluation of solid fusion for fracture and clinical outcomes. The institutional review board approved the study protocol (PC23RISI0066), ensuring compliance with the ethical guidelines of the 1975 Helsinki Declaration.

### 2.2. Surgical Techniques

After general anesthesia, patients were placed on the four-poster frame with the abdomen suspended to avoid excessive epidural bleeding and restore the collapsed vertebral height by extension of the thoracolumbar junction [15]. After a midline longitudinal skin incision, periosteal dissection along the paravertebral muscles was carried out to expose the lamina and facets. After identifying the entry points for screw insertion, under-tapping by 1 mm was performed to prevent screw loosening. As tapping was loose, 2 or 3 cc of cement augmentation at the upper and lower instrumented vertebra was carried out before the screw insertion. Then, a pre-bent rod on one side was placed into the screws. After the cap tightening of screws below the fractures, distraction between the screw above the fracture and the rod holder was carried out for maximal reduction of the fracture site. After one-side rod fixation, serial pedicle dilation was carried out to insert the funnel for BMP impaction. A hydroxyapatite-based alloplastic bony substitute with 1 mg/mL of *E. coli*-derived rhBMP-2 (NOVOSIS^®^, CGBio Inc., Seongnam, Republic of Korea) and 15 cc of the allogenous bone chip were impacted through the funnel (Figure 1). After BMP impaction, bone wax was sealed for entry of the pedicle to prevent the leakage of BMP. Then, the pre-bent rod on the same side was placed into the screws and fixated with distraction for the fracture site. The wound was closed with the insertion of a drain. At postoperative 1 or 2 days, patients were allowed to ambulate with thoracolumbosacral orthosis.

### 2.3. Radiologic and Clinical Parameters

Simple plain radiographs were obtained once a week during the first month after surgery, once a month until the third month, and once every three months until the union was confirmed. Computed tomography was carried out post-operatively for six months and one year. Without definite evidence of a vertebral cleft, trabecular continuity or bridging of bone around the cleft was regarded as a radiographic union [16]. Two spine surgeons assessed the radiographic union (each surgeon evaluated the union independently, and another surgeon assessed when the two surgeons differed). As radiologic parameters, local kyphotic angle (LKA), anterior vertebral height (AVH), and vertebral wedge angle (VWA) were measured on neutral lateral radiographs (Figure 2) [17].

Clinical outcomes included 10-point Numeric Rating Scale (NRS) scores for back and leg pain and Oswestry Disability Index (ODI) scores as patient-reported outcomes. To examine the quality of life, EuroQol-5 Dimension (EQ-5D), composed of mobility, self-care, usual activities, pain/discomfort, and anxiety/depression, was assessed. The ED-5D was converted into EQ-5D utility index scores anchored at 0 for death and 1 for perfect health. The American Spinal Injury Association (ASIA) impairment scale was assessed to range from A to E to assess the degree of neurologic deficits (A = complete, B, C, D = incomplete, E = normal).

### 2.4. Statistical Analysis

Radiologic parameters, presented as means and standard deviations, were compared between pre-operative, post-operative, and last follow-up using the Wilcoxon signed-rank test. Clinical parameters, presented as means and standard deviations, were compared between pre-operative and last follow-up using the Wilcoxon signed-rank test. Statistical analyses were performed using SPSS software (IBM SPSS Statistics for Windows, Version 24.0, IBM Corp., Armonk, NY, USA) with a level of statistical significance of 0.05.

## 3. Results

### 3.1. Patient Demographics

Ten patients undergoing posterior spinal fusion using *E. coli*-derived rhBMP-2 with more than one year of follow-up were assessed in this study. Patient demographics are presented in Table 1. The average T-score of bone mineral density was −1.6 ± 1.7, and the fracture levels were primarily thoracolumbar lesions. The average fusion level was 4.1 ± 1.0, and four patients (40%) underwent cement augmentation. The average operation time was 184.6 ± 70.0 min, and the estimated bleeding loss was 315.0 ± 152.8 mL. The transfusion was carried out for three patients with a mean of 306.7 ± 123.6 mL.

Details of patients undergoing surgery are represented in Table 2. Anabolic agents, such as teriparatide and romosozumab, were used in nine (90%) patients with an average use of 5.2 ± 3.9 months. The average time of radiographic union was 99.9 ± 45.4 (62–192) days, and the mean follow-up period was 15.5 ± 3.7 months.

### 3.2. Radiographic Outcomes

Radiographic outcomes are presented in Table 3. LKA was significantly corrected from 9.9 ± 17.9° to −0.8 ± 16.5° after the surgery and 3.8 ± 18.6° at the last follow-up without significance. AVH also significantly increased from 19.1 ± 8.3 mm to 28.7 ± 3.2 mm after the surgery and 26.4 ± 4.9 mm at the last follow-up (all *p*-values < 0.05 compared to pre-operative height). In addition, VWA was significantly decreased from 12.3 ± 8.3° to 2.5 ± 4.6° after the surgery and 4.7 ± 5.6° at the last follow-up (all *p*-values < 0.05 compared to pre-operative angle).

### 3.3. Clinical Outcomes

Clinical outcomes are presented in Table 4. NRS for back pain was significantly decreased between pre-operatively and at the last follow-up (8.6 ± 1.8 vs. 3.7 ± 1.8, *p* < 0.001). NRS for leg pain was decreased without significance (3.4 ± 3.7 vs. 2.4 ± 2.3, *p* = 0.393). The patient also reported significantly lower disability at the last follow-up (67.8 ± 17.5% vs. 36.0 ± 13.3, *p* = 0.002). In addition, the EQ-5D utility index score at the last follow-up was significantly greater, meaning better quality of life (0.21 ± 0.28 vs. 0.68 ± 0.18, *p* = 0.001).

Regarding the neurologic outcomes, four patients had neurologic deficits before the surgery. One patient had an ASIA impairment scale of C, and three patients showed D. At the last follow-up, three patients with grade D recovered after surgery. However, one patient with grade C improved to grade D and could ambulate with a walker. Any adverse event related to using *E. coli*-derived rhBMP-2, such as heterotopic ossification and seroma formation, was not observed in this cohort.

### 3.4. Representative Case

A 75-year-old female patient was diagnosed with a D11 compression fracture after falling from her height (Patient 7 in Table 2). She underwent intravenous antibiotics for a prevertebral abscess. One month later, she presented with dorsal back pain and lower leg weakness. As kyphosis progressed with neurologic deficits, surgery was decided. Only posterior surgery applying *E. coli*-derived rhBMP-2 to the reduced fractured body was carried out instead of combined anterior-posterior surgery. After the surgery, kyphosis and collapsed vertebra were corrected. Bony union was observed three months post operation and maintained at the last follow-up without loss of correction (Figure 3).

## 4. Discussion

Recent meta-analysis for rhBMP-2 in lumbar fusion has been reported to show a higher fusion success rate (odds ratio [OR] = 3.79) and lower re-operation rate (OR = 0.59) compared to autogenous iliac bone grafts [18]. *E. coli*-derived rhBMP-2 has also revealed the efficacy of better efficacies of bony fusion compared to autogenous iliac bone graft. First, Cho et al. reported that the fusion rate of posterolateral fusion using *E. coli*-derived rhBMP-2 was 100% compared with 90.2% of autogenous iliac bone grafts at 12 weeks [11]. Son et al. also reported that 0.3 mg of rhBMP-2 with hydroxyapatite could achieve a successful fusion rate of 95.2% in anterior cervical discectomy and fusion. Additionally, accelerated fusion was observed three months after surgery in 15 of 21 segments (71.4%) [12]. Although the nonunion of lumbosacral junction in adult spinal deformity has been reported high in previous studies, 3 mg of *E. coli*-derived rhBMP-2 achieved 100% of anterior lumbar interbody fusion twelve months after the surgery [13].

In this context, this study hypothesized that applying rhBMP-2 to the fractured vertebral body could accelerate bony fusion. Finally, if earlier solid fusion could be possible, posterior surgery alone would be an effective surgical option instead of combined anterior-posterior surgery. This hypothesis was the start of this case series. Although allogenous bone graft impaction has been carried out previously, the clinical efficacy of rhBMP-2 for unstable spinal fractures has not been reported.

In this study, the average period of fracture healing was 99.9 ± 45.4 (62–192) days. This result was consistent with the previous study. In single-level lumbar or lumbosacral posterolateral fusion, Cho et al. reported that the CT-based fusion rate in the *E. coli*-derived rhBMP-2 group was 100.0% (41/41) at 12 weeks, while 90.2% (46/51) in the autogenous iliac bone graft group [11]. Similarly, upper extremity fractures treated with rhBMP-2 reached radiographic union in 117 days [19]. Even in long bone non-unions, rhBMP-2 promoted faster bone healing (HR = 2.78; 95% CI 1.4–5.6, *p* < 0.001) and higher union rates (89% vs. 47%; *p* < 0.001) compared to no-BMP group [20]. Although previous studies reported no advantages of rhBMP-2 on fracture healing, recent studies have reported better fracture healing than control or even autogenous iliac bone graft [21,22].

The radiographic pattern of fracture healing, as seen in Figure 3B, was also interesting in the present study. Even if empty space in the fractured body remains, a solid bony union at the fracture margin was achieved. This finding supports our hypothesis that accelerated fracture healing could support anterior load sharing and provide spinal stability without anterior surgery. This study revealed that the rhBMP-2 could enhance the healing process of the fractured vertebral body. Regarding the mechanism of rhBMP-2 for fracture healing, Liu et al. reported that rhBMP-2 accelerated the migration of bone marrow mesenchymal stem cells via the CDC42/PAK1/LIMK1 pathway and enhanced the fracture healing process [21]. Another study demonstrated the BMP-2-dependent fracture healing through tight control of chemokine C-X-C motif-ligand-12 (CXCL12) expression [23]. However, the mechanism of fracture healing remains to be discussed in further trials.

The relationship between BMP-2 and anabolic agents is also interesting. In this study, nine patients used anabolic agents for an average of 5.2 ± 3.9 months. In a rat model of rhBMP-2-induced spinal fusion, Morimoto et al. reported that teriparatide significantly increased fusion rates and improved the quality of bone formation [24]. The authors suggested that the combined administration of rhBMP-2 and teriparatide might lead to efficient spinal fusion. In recent clinical studies, using anabolic agents such as teriparatide could be helpful for fracture healing [1,25]. Ebata et al. reported that teriparatide promoted bone formation at the fusion site and decreased bone resorption during the early postoperative period, as indicated by bone turnover markers [26]. Moreover, teriparatide can significantly reduce the risk of additional fractures after a spinal fracture [27]. Since the association between romosozumab and BMP-2 has not been reported, further studies are needed to evaluate the relationship between anabolic agents and BMP-2.

Successful fusion at a relatively early phase might decrease the correction loss of the fracture site. Our previous study reported that cement augmentation for Kummell’s disease with long-segmental posterior fusion was effective [28]. Although the previous study included patients with Kummell’s disease, the long-instrumented fusion of more than five levels to achieve posterior fusion was needed, and a significant loss of correction was observed at the last follow-up. However, this study revealed maintenance of kyphosis correction at the last follow-up without progressive loss of correction. Maintenance of restored sagittal alignment without sagittal imbalance can prevent the further collapse of the fracture and result in better outcomes through decreasing back pain due to sagittal deformity and prevention of neurologic deficits [29].

Regarding safety issues, rhBMP-2-related complications for spinal surgery have been reported. Those adverse events included postoperative radiculitis, ectopic bone formation, vertebral osteolysis/edema, dysphagia and neck swelling, hematoma formation, and wound healing complications [4,30]. Those complications are associated with a high risk of occurrence dose-dependently. This study used several measures to prevent potential complications and adverse events. First, rhBMP-2 was applied only to the fractured body and sealed with bone wax to prevent leakage. Second, potential rhBMP-2-related adverse events, such as fever, tissue swelling, and radiculitis, were closely observed after the surgery. However, no adverse events were reported in this study.

Surgical indications for the use of rhBMP-2 can be as follows. First, acute burst fractures with severe vertebral body collapse could be a better indication for this procedure. After distraction, the vertebral body could be restored to vertebral height before the injury. Restoration of the fractured body might provide better spinal alignment and clinical outcomes. Second, Kummell’s disease could be another indication of this procedure. Intra-vertebral cleft means the instability of a fractured body and the possibility of restoration of the fracture [31]. This technique could be carried out only in the posterior approach. Especially for elderly patients with co-morbidities, this surgery can be effective and safer regarding shorter operation time and less bleeding compared to the combined anterior-posterior surgery [32]. However, a solid union of the fractured body is a surgical contraindication. Because restoring a fractured body is not impossible, an osteotomy can be needed for patients with solid fracture unions.

Meanwhile, this technique could also be applied to minimally invasive spine surgery (MISS). For young patients considering a short level of fusion or implant removal, applying the rhBMP-2 could be possible via percutaneous maneuver after fixation of percutaneous pedicle screws. It would be better because MISS for unstable thoracolumbar burst fracture showed significant loss of postoperative back pain, operation time, and blood loss compared to open instrumentation surgery [33,34].

This study has some limitations. First, the number of included cohorts was small. We started this type of surgery in 2021. However, we included the patients with a minimum one-year follow-up to evaluate the solid bony union and clinical outcome. Second, *E. coli*-derived rhBMP-2 has been approved for posterolateral fusion. However, *E. coli*-derived rhBMP-2 has been used for other types of spinal surgery [11,12,13]. No adverse events related to rhBMP-2 was not observed in this study. Third, this study was a case series; this surgical technique was not compared with control groups or other surgical strategies. The efficacies of this pilot study should be re-assessed in further trials. Finally, this study included heterogeneous cohorts with different diagnoses (traumatic and osteoporotic), the use of anabolic agents, and cement augmentation, even if all fractures were unstable fractures requiring surgical interventions. Thus, further trials with a large number of cohorts are needed to validate the efficacies of this technique by comparing it with other surgical techniques and evaluating the results according to specific surgical indications.

Despite these limitations, this study suggested a new surgical option using rhBMP-2 in unstable spinal fractures. In addition, anabolic agents could help achieve earlier solid fusion. In advancing biomaterials and pharmaceuticals, surgical techniques could be changed for better clinical outcomes.

## 5. Conclusions

Combined with the anabolic agents, the application of *E. coli*-derived rhBMP-2 to the fractured vertebral body could achieve earlier solid fusion around 100 days without any adverse events, which might decrease the loss of correction and obtain better clinical outcomes. However, further comparative studies with a large number of patients are needed to validate the advantage of this technique over other techniques and identify detailed surgical indications.

## Figures and Tables

**Figure 1 bioengineering-10-01114-f001:**
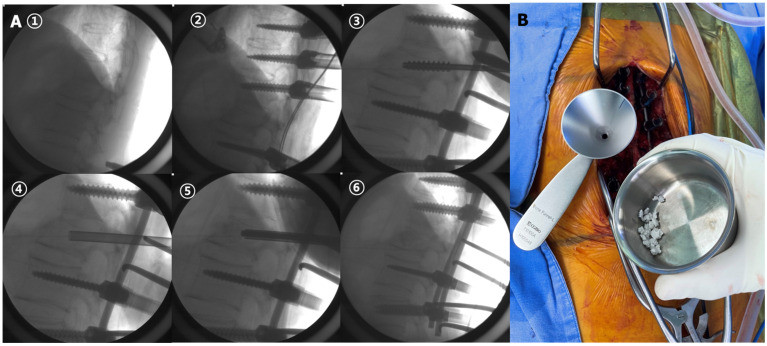
(**A**) Surgical steps: ① postural reduction of fracture using four-poster frame, ② pedicle screws insertion, ③ serial pedicle dilation at another side after rod fixation with distraction on one side, ④ bone funnel insertion, ⑤ rhBMP-2 and allogenous bone chip impaction, ⑥ final rod fixation; (**B**) bone funnel inserted into fractured body and hydroxyapatite-based alloplastic bony substitute with 1 mg of *E. coli*-derived rhBMP-2.

**Figure 2 bioengineering-10-01114-f002:**
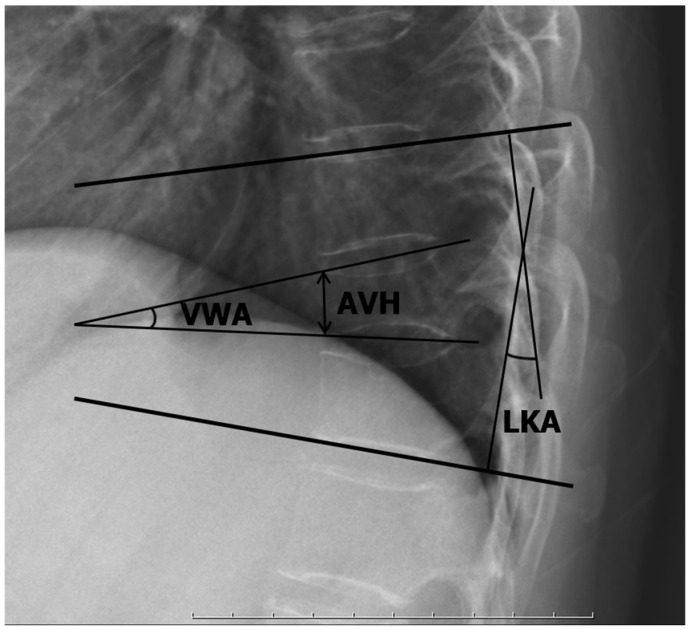
Radiologic parameters; local kyphotic angle (LKA), anterior vertebral height (AVH), and vertebral wedge angle (VWA).

**Figure 3 bioengineering-10-01114-f003:**
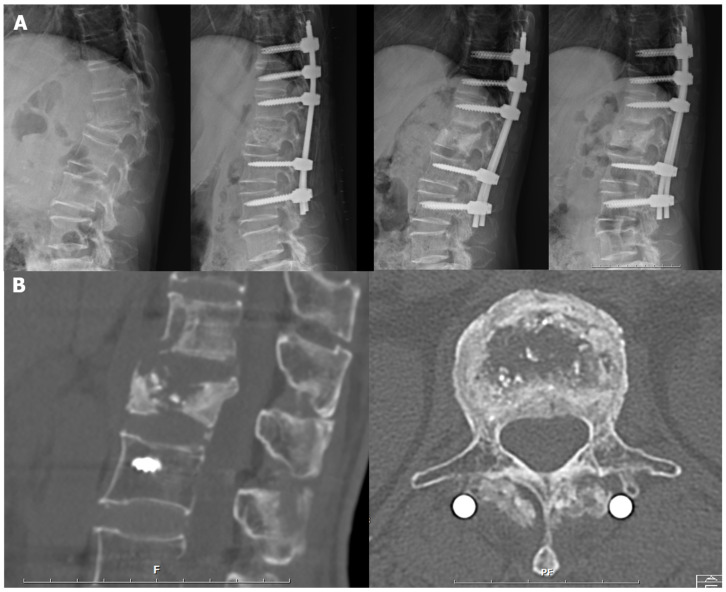
A representative case: (**A**) serial plain radiographs preoperatively, postoperatively, postoperatively three months and one year, (**B**) sagittal and axial cuts of computed tomography at the level of the fractured body postoperatively one year.

**Table 1 bioengineering-10-01114-t001:** Demographics of included patients.

Variables	Number (Percent)
**Number of patients**	10
**Sex (M:F)**	5:5
**Age (years)**	71.7 ± 7.9
**Smoking**	2 (20%)
**Bone mineral density (T-score)**	−1.6 ± 1.7
**Fractured level**	D11: 1 D12: 3 L1: 4 L3: 1 L5: 1
**Level of fusion**	4.1 ± 1.0
**Cement augmentation**	4 (40%)
**Operation time (minutes)**	184.6 ± 70.0
**Bleeding (mL)**	315.0 ± 152.8
**ASA (1:2:3:4)**	0:9:1:0

Note: M—male; F—female; ASA—American society of anesthesiologists.

**Table 2 bioengineering-10-01114-t002:** Details of patients undergoing spinal surgery.

Patients	Age/Sex	Diagnosis	Fused Level	Anabolic Agent	Radiographic Healing (Days)	Follow-Up Period (Months)
**1**	71/M	L5 unstable burst fracture	L3-S1 with S2AI screws	Teriparatide (3 months)	96	17
**2**	75/M	D12 chalky stick fracture Underlying AS	D10-L2 (D10, L2 cement augmentation)	Teriparatide (6 months)	173	13
**3**	81/F	D12 chalky stick fracture Underlying AS	D9-L2 (L2 cement augmentation)	Teriparatide (8 months)	100	23
**4**	75/F	L1 unstable burst fracture	D11-L2 (D11,12,L2 cement augmentation)	Teriparatide (1 month)	68	18
**5**	68/M	L3 flexion-distraction injury	L1–4	Teriparatide (1 month)	83	18
**6**	70/M	L1 unstable burst fracture	D11–L2	Teriparatide (5 months)	67	17
**7**	78/F	L1 unstable burst fracture Combined prevertebral abscess	D10–L3	Teriparatide (1 month) → Romosozumab (7 months)	84	13
**8**	74/F	D12 unstable burst fracture	D10–L2	Romosozumab (12 months)	192	12
**9**	52/M	L1 unstable burst fracture	D11–L3	None	74	12
**10**	73/F	D11 unstable burst fracture	D8–L2 (L2 cement augmentation)	Teriparatide (2 months) → Romosozumab (6 months)	62	12
**Total**	-	-	4.1 ± 1.0 levels	5.2 ± 3.9 months	99.9 ± 45.4	15.5 ± 3.7

Note: M—male; F—female; AS—ankylosing spondylitis.

**Table 3 bioengineering-10-01114-t003:** Radiologic parameters at pre-operative, post-operative and last follow-up.

	LKA	AVH	VWA
**Pre-operative**	9.9 ± 17.9	19.1 ± 8.3	12.3 ± 8.3
**Post-operative**	−0.8 ± 16.5	28.7 ± 3.2	2.5 ± 4.6
***p*-value ***	0.002	0.001	0.001
**Last follow-up**	3.8 ± 18.6	26.4 ± 4.9	4.7 ± 5.6
***p*-value ****	0.072	0.006	0.006

Note: LKA—local kyphotic angle; AVH—anterior vertebral height; VWA—vertebral wedge angle. * means *p*-value between pre-operative and post-operative. ** means *p*-value between pre-operative and last-follow-up.

**Table 4 bioengineering-10-01114-t004:** Clinical outcomes at pre-operative and last follow-up.

	Pre-Operative	Last Follow-Up	*p*-Value
**NRS (back pain)**	8.6 ± 1.8	3.7 ± 1.8	<0.001
**NRS (leg pain)**	3.4 ± 3.7	2.4 ± 2.3	0.393
**ODI**	67.8 ± 17.5	36.0 ± 13.3	0.002
**EQ-5D index**	0.21 ± 0.28	0.68 ± 0.18	0.001
**ASIA impairment scale**			
**A**	0	0	-
**B**	0	0	
**C**	1	0	
**D**	3	1	
**E**	6	9	

Note: NRS—numeric rating scale; ODI—Oswestry disability index; ASIA—American spinal injury association.

## Data Availability

Original data will be made available upon reasonable request.

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
