# Peer review of "Application of Escherichia coli-Derived Recombinant Human Bone Morphogenic Protein-2 to Unstable Spinal Fractures"

_bioengineering, 2023, doi:10.3390/bioengineering10101114_

Round 1
Reviewer 1 Report
The study by Kim et al. delves into the application of recombinant Human Bone Morphogenic Protein-2 (rhBMP-2) for the treatment of unstable spinal fractures. The subject matter is particularly intriguing given the FDA approval status of rhBMP-2 for procedures such as spinal fusion, tibial shaft repair, and maxillary sinus reconstructive surgery.
Nonetheless, the current version of the manuscript remains unsuitable for publication primarily due to its organizational flaws.
General comments
· Please italicize the species name, Escherichia coli.
· The introduction lacks depth and coherence. The authors should expand this section to provide better context.
· Delve deeper into the physiological role of Bone Morphogenic Protein-2 (BMP-2), particularly in development and bone homeostasis.
· P.1, LL. 40-41: The statement regarding BMP-2's potential post-translational modifications is ambiguous. Kindly elucidate.
· Stemming from the previous point, the authors should discuss the possible limitations of using a protein synthesized in E. coli versus one produced using a mammalian expression system.
· The potential side effects of administering recombinant proteins warrant a thorough discussion. The methodology used to monitor, prevent, or manage these side effects should be clarified.
· Continuously referring to the recombinant BMP-2 as "E-coli derived rhBMP-2" seems redundant. Using "rhBMP-2" should suffice.
· The methods section should be expanded to detail the patient recruitment process, including explicit inclusion and exclusion criteria.
· The absence of a control group is a glaring shortcoming. An explanation or justification for this oversight is essential.
· In Table 2, font size and style inconsistencies need rectification.
· LL 201-202: This statement needs to be bolstered with appropriate references.
· The authors should engage in a more detailed discussion concerning the employment of recombinant proteins in spinal surgeries.
· P.9, LL268-269: More comprehensive details about the various spinal procedures where rhBMP-2 finds utility would enhance understanding.
The manuscript lacks organization and needs a significant reconstruction.
Author Response
Response to Reviewer 1 Comments
The study by Kim et al. delves into the application of recombinant Human Bone Morphogenic Protein-2 (rhBMP-2) for the treatment of unstable spinal fractures. The subject matter is particularly intriguing given the FDA approval status of rhBMP-2 for procedures such as spinal fusion, tibial shaft repair, and maxillary sinus reconstructive surgery.
Nonetheless, the current version of the manuscript remains unsuitable for publication primarily due to its organizational flaws.
Authors’ response: Thank you for your comment. We tried to revise the manuscript to make it a better study, considering your valuable comments.
Point 1: Please italicize the species name, Escherichia coli.
Response 1: We agree with your comment. We corrected the name of the bacteria in italics as your comment.
Point 2: The introduction lacks depth and coherence. The authors should expand this section to provide better context.
Response 2: Thank you for your comment. We revised the introduction considering your comments for Bone Morphogenic Protein-2 (BMP-2).
Point 3: Delve deeper into the physiological role of Bone Morphogenic Protein-2 (BMP-2), particularly in development and bone homeostasis.
Response 3: Thank you for your comment. As you commented, we suggested the physiologic role and development of BMP-2.
Point 4: P.1, LL. 40-41: The statement regarding BMP-2's potential post-translational modifications is ambiguous. Kindly elucidate.
Response 4: Thank your for your comment. Post-translation problems mean incomplete monomer processing. We revised the sentence and added the reference.
Point 5: Stemming from the previous point, the authors should discuss the possible limitations of using a protein synthesized in E. coli versus one produced using a mammalian expression system.
Response 5: Thank you for your comment. In the introduction, we added the limitations and advantages of E.coli-derived rhBMP-2 compared to Chinese Hamster Ovary cell rhBMP-2.
Point 6: The potential side effects of administering recombinant proteins warrant a thorough discussion. The methodology used to monitor, prevent, or manage these side effects should be clarified.
Response 6: We agree with your comment. In the discussion section, we added the potential adverse events related to rhBMP-2 and suggested measures to prevent the complications.
Point 7: Continuously referring to the recombinant BMP-2 as "E-coli derived rhBMP-2" seems redundant. Using "rhBMP-2" should suffice.
Response 7: Thank you for your comment. We changed the E.coli-dervied rhBMP-2 to the rhBMP-2 unless necessary.
Point 8: The methods section should be expanded to detail the patient recruitment process, including explicit inclusion and exclusion criteria.
Response 8: Thank you for your commment. We revised the material section to suggest the inclusion and exclusion criteria details as you commented.
Point 9: The absence of a control group is a glaring shortcoming. An explanation or justification for this oversight is essential.
Response 9: We agree with your comment. This study was a case series; this surgical technique was not compared with control groups or other surgical strategies. We suggested this as the limitation of this study and the necessity of further trials.
Point 10: In Table 2, font size and style inconsistencies need rectification.
Response 10: Thank you for your comment. As you commented, we increased the font size and revised the table with the same style.
Point 11: LL 201-202: This statement needs to be bolstered with appropriate references..
Response 11: Thank you for your comment. We added the meta-analysis for rhBMP-2 compared to autogenous iliac bone grafts in lumbar fusion.
Point 12: The authors should engage in a more detailed discussion concerning the employment of recombinant proteins in spinal surgeries.
Response 12: Thank you for your comment. We suggested the use of rhBMP-2 for various spinal surgeries in the discussion section.
Point 13: P.9, LL268-269: More comprehensive details about the various spinal procedures where rhBMP-2 finds utility would enhance understanding.
Response 13: Thank you for your comment. As mentioned earlier, we suggested using rhBMP-2 for various spinal surgeries in the discussion section.

Reviewer 2 Report
Dear author
This is good paper but need some changes before to be accepted.
This paper could be improved it. Please see my suggestions from below:
Please prepare better aim
Please remove from materials if is not about this
Surgical indications for unstable spinal fractures were mainly mechanical instability of fractures 62 and neurologic involvement [1].
Please remove all references from materials if is not about materials
Figure 2. could you include a scale in this image?
Table 2. too less number of patients!!
Increase size of fonts from table
Figure 3. a scale inside of images could be better
please improve language
Author Response
Response to Reviewer 2 Comments
This is good paper but need some changes before to be accepted.
This paper could be improved it. Please see my suggestions from below:
Authors’ response: Thank you for your positive comment. We tried to revise the manuscript to make it a better study. Regarding the English presentation, we went through two English correction processes. First, an English correction program using Grammarly was done. Second, the manuscript was corrected again by a fluent English colleague.
Point 1: Please prepare better aim.
Response 1: Thank you for your comment. This study evaluated the efficacy and safety of E.coli-derived rhBMP-2 for unstable vertebral fractures for the first time. We revised the paragraph to emphasize the purpose of this study as your comment.
Point 2: Please remove from materials if is not about this
Surgical indications for unstable spinal fractures were mainly mechanical instability of fractures and neurologic involvement [1].
Response 2: Thank you for your comment. We revised the methods sections for inclusion and exclusion criteria as your comment.
Point 3: Please remove all references from materials if is not about materials.
Response 3: Thank you for your comment. As you commented, we removed references from materials if they are not about materials.
Point 4: Figure 2. could you include a scale in this image?
Response 4: Thank you for your comment. We changed Figure 2 and added the scale bar of 5mm at the bottom of the figure as your comment.
Point 5: Table 2. too less number of patients!!
Response 5: We agree with your comment. We started this type of surgery in 2021 and included the patients with a minimum one-year follow-up to evaluate the solid bony union and clinical outcome. We suggested the small number of patients as the limitation and necessity of further trials with a large number of cohorts.
Point 6: Increase size of fonts from table.
Response 6: Thank you for your comment. We increased the size of the fonts from the tables.
Point 7: Figure 3. a scale inside of images could be better.
Response 7: Thank you for your comment. We added the scale bar of 5mm at the bottom of the figure as your comment.

Reviewer 3 Report
The work presented for publication is in the nature of a description of clinical management, including studies of a medical experimental nature. Its basic assumptions are fairly well defined and justified in the literature review conducted as part of the introduction chapter.
The authors refer to previous work using recombinant human bone morphogenetic proein-2 (rhBMP-2) in osteoinduction.
The work has been well-documented which makes it possible to accurately trace the medical management applied to the patients. This is all the more important in view of the very interesting results indicating great final success.
However, the authors, during the preparation of the paper, did not avoid minor inaccuracies or errors of an editorial nature, the completion or correction of which will allow acceptance of the work.
1. E.coli is an abbreviation for the Latin species name of bacteria and should always be written in italics.
2. The authors of the paper use "E-coli derived rhBMP-2" in the initial part of the article, while the correct name should be "E.coli derived rhBMP-2", which can also be checked on the website of the preparation's manufacturer (https://www.cgbio.co.kr/en/product/bone-spinal/novosis).
3. Some of the statistical data given by the authors should be revised, because giving the value plus the standard deviation gives a nonsensical picture from a statistical point of view - as an example, the value of transfusion volume in ml can be shown in Table 1 - "92.0 ml with a statistical deviation of 164.4 ml" would have to mean that it is possible to "collect 72 ml of blood from the patient", which is an obvious absurdity and shows that this parameter should not be averaged in this way.
A similar situation occurs in the case of Table 3, also in this case giving the mean plus the standard deviation is a statistical oversimplification that does not give a correct picture of the data obtained, even more so when a very small p-value range is given.
Finally, I would like to emphasize that these comments minimally diminish my high assessment of the work.
Author Response
Response to Reviewer 3 Comments
The work presented for publication is in the nature of a description of clinical management, including studies of a medical experimental nature. Its basic assumptions are fairly well defined and justified in the literature review conducted as part of the introduction chapter.
The authors refer to previous work using recombinant human bone morphogenetic proein-2 (rhBMP-2) in osteoinduction.
The work has been well-documented which makes it possible to accurately trace the medical management applied to the patients. This is all the more important in view of the very interesting results indicating great final success.
However, the authors, during the preparation of the paper, did not avoid minor inaccuracies or errors of an editorial nature, the completion or correction of which will allow acceptance of the work.
Authors’ response: Thank you for your comment. Considering your valuable comments, we tried to revise the manuscript to make it a better study.
Point 1: E.coli is an abbreviation for the Latin species name of bacteria and should always be written in italics.
Response 1: We agree with your comment. We corrected the name of the bacteria in italics as your comment.
Point 2: The authors of the paper use "E-coli derived rhBMP-2" in the initial part of the article, while the correct name should be "E.coli derived rhBMP-2", which can also be checked on the website of the preparation's manufacturer (https://www.cgbio.co.kr/en/product/bone-spinal/novosis).
Response 2: Thank you for your comment. As you commented, we revised the name to E.coli-derived rhBMP-2.
Point 3: Some of the statistical data given by the authors should be revised, because giving the value plus the standard deviation gives a nonsensical picture from a statistical point of view - as an example, the value of transfusion volume in ml can be shown in Table 1 - "92.0 ml with a statistical deviation of 164.4 ml" would have to mean that it is possible to "collect 72 ml of blood from the patient", which is an obvious absurdity and shows that this parameter should not be averaged in this way.
A similar situation occurs in the case of Table 3, also in this case giving the mean plus the standard deviation is a statistical oversimplification that does not give a correct picture of the data obtained, even more so when a very small p-value range is given.
Response 3: We agree with your comment. For the transfusion three patients took the transfusion, while seven patients did not. As you commented, we revised the manuscript to suggest the mean and standard variation of three patients undergoing transfusion. In Table 3, the local kyphotic angle could be positive or negative according to kyphosis or lordosis. Two patients underwent spinal surgery for lumbar fractures. The kyphotic angle in two patients was negative, while others were positive. In this regard, we suggested the standard error values for the local kyphotic angle.

Round 2
Reviewer 1 Report
I have no further comments. The manuscript can be accepted in its current form.